# Perceptual Attacks of No-Reference Image Quality Models with Human-in-the-Loop

**Weixia Zhang**[1]     **Dingquan Li**[2]     **Xiongkuo Min**[1]     **Guangtao Zhai**[1]

**Guodong Guo**[3]          **Xiaokang Yang**[1]          **Kede Ma**[4*]

[1] MoE Key Lab of Artificial Intelligence, AI Institute, Shanghai Jiao Tong University
[2] Network Intelligence Research Department, Peng Cheng Laboratory
[3] Department of Computer Science and Electrical Engineering, West Virginia University
[4] Department of Computer Science, City University of Hong Kong

{zwx8981, minxiongkuo, zhaiguangtao, xkyang}@sjtu.edu.cn, lidq01@pcl.ac.cn
guodong.guo@mail.wvu.edu, kede.ma@cityu.edu.hk

## Abstract

No-reference image quality assessment (NR-IQA) aims to quantify how humans perceive visual distortions of digital images without access to their undistorted references. NR-IQA models are extensively studied in computational vision, and are widely used for performance evaluation and perceptual optimization of man-made vision systems. Here we make one of the first attempts to examine the perceptual robustness of NR-IQA models. Under a Lagrangian formulation, we identify insightful connections of the proposed perceptual attack to previous beautiful ideas in computer vision and machine learning. We test one knowledge-driven and three data-driven NR-IQA methods under four full-reference IQA models (as approximations to human perception of just-noticeable differences). Through carefully designed psychophysical experiments, we find that all four NR-IQA models are vulnerable to the proposed perceptual attack. More interestingly, we observe that the generated counterexamples are not transferable, manifesting themselves as distinct design flows of respective NR-IQA methods. Source code are available at https://github.com/zwx8981/PerceptualAttack_BIQA.

## 1   Introduction

Over the past decade, deep neural networks (DNNs) have revolutionized a wide range of computer vision and machine learning applications. For example, ResNet [1] reduced the object recognition error rate on ImageNet [2] to $3.56\%$, surpassing the performance of informed humans. A similar story is written in the field of image quality assessment (IQA), where DNN-based IQA models [3, 4, 5, 6] exhibit very high correlations with human opinions of perceptual quality on various subject-rated datasets [7, 8, 9, 5]. IQA models can be roughly classified into full-reference IQA (FR-IQA) and no-reference IQA (NR-IQA) ones depending on the availability of the pristine undistorted image as reference. Representative FR-IQA methods include the Minkowski distance (*i.e.*, the $\ell_p$-norm induced metric), the structural similarity (SSIM) index [10], the learned perceptual image patch similarity (LPIPS) method [3], and the deep image structure and texture similarity (DISTS) model [4], which are widely used for measuring signal fidelity and quality in various vision applications. NR-IQA

---

[*]Corresponding author.

36th Conference on Neural Information Processing Systems (NeurIPS 2022).

models [11] are extensively studied in computational vision, which mimic the human ability to judge the perceptual quality of a test image without comparison to any reference image. NR-IQA plays an indispensable role in the design and optimization of real-world image processing algorithms.

Despite the remarkable achievements of DNN-based models, recent work has identified their vulnerability to adversarial perturbations [12, 13]. For example, in natural image classification, a visually indistinguishable perturbation added to a natural image would mislead "top-performing" classifiers. This imperceptible perturbation is called below the just-noticeable difference (JND) of human perception in psychophysics, and is less relevant in image classification [14]. This is because for the majority of practical adversarial attacks [15] considered in image classification (*e.g.*, the $\ell_\infty$-norm constrained attack), the allowable perturbations, even if above JNDs, do not alter the image semantics: they are label-preserving. In the context of IQA, it becomes highly nontrivial to craft label-preserving attacks for two main reasons. First, for adversarial perturbations that are above JNDs, they are highly likely to be perceived as some form of visual distortions that lead to quality degradation (see Fig. 1). Second, computational prediction of JNDs for natural images [16] remains an open research problem as it depends on the combination of the image content and the perturbation type, constrained by the psychophysical experimental conditions (*e.g.*, the maximum time of visual inspection).

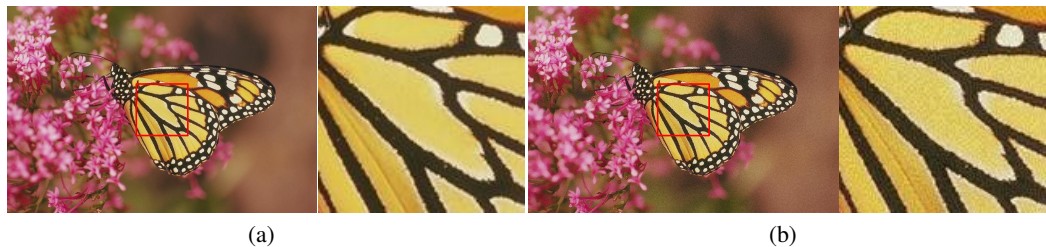

(a)          (b)

Figure 1: Visual comparison of **(a)** a natural photographic image and **(b)** a computer-generated example from (a) by attacking BRISQUE [17] using the projected gradient method, under an $\ell_\infty$-norm constraint with a radius of $8/255$. It is clear that the semantic information of the image remains intact, while its perceptual quality degrades due to the introduced "mosquito" noise.

In this work, we take initial steps to examine the perceptual robustness of NR-IQA models. Our main contributions are threefold.

- We propose a two-step perceptual attack for NR-IQA methods with human-in-the-loop. First, we define the objective of the perceptual attack as a Lagrangian function of an FR-IQA model (as the "perceptual" constraint) and the NR-IQA model (to be examined). By varying the Lagrange multiplier, we generate a series of perturbed images of different distortion visibility. Second, we ask human observers in a well-controlled psychophysical experiment to determine whether the perturbation of each image is discriminable. The output of our perceptual attack is a computer-generated counterexample, that is below the JND while leading to the most significant change in quality prediction.
- We draw connections between the proposed perceptual attack and conventional wisdom in literature, including the Carlini-Wagner attack [18] in computer vision, maximum a posterior (MAP) estimation in statistical signal processing, maximum differentiation (MAD) competition [19] in computational vision, and eigen-distortion analysis [20] in computational neuroscience.
- We conduct an extensive experiment to examine four NR-IQA models, the knowledge-driven BRISQUE [17], the shallow learning-based CORNIA [21], as well as the deep learning-based Ma19 [22] and UNIQUE [6] under four FR-IQA models, the Chebyshev distance (*i.e.*, the $\ell_\infty$-norm induced metric), SSIM, LPIPS, and DISTS (as approximations to human perception of JNDs). We arrive at several important observations, among which the most interesting one is that the proposed perceptual attack succeeds in fooling all four NR-IQA models, but the generated counterexamples are not transferable, manifesting themselves as distinct design flows of respective NR-IQA methods.

# 2 Related Work

In this section, we give a review of NR-IQA models, and summarize representative adversarial attacks in classification, and discuss them in a broader context of "analysis by synthesis" [23, 24].

## 2.1 NR-IQA Models

Early NR-IQA models were designed to deal with specific distortion types, *e.g.*, JPEG compression [25] and JPEG2000 compression [26]. In the past decade, research on general-purpose NR-IQA become popular with the proposal of a variety of quality-aware features based on natural scene statistics (NSS) [17, 11, 27]. Unsupervised feature learning such as codebook construction [21] was also explored. Since the work of [28], DNN-based methods began to revolutionize the field of NR-IQA. Towards developing more accurate NR-IQA models, recent research includes joint learning from multiple databases [6], active learning for improved generalizability [29], patch-to-picture learning for local quality prediction [30], and continual learning for handling streaming visual data [31].

## 2.2 Adversarial Attacks in Classification

Machine learning models are, for a long time, known to be vulnerable to adversarial examples [12, 32, 33, 34]: data samples that have been modified very slightly but enough to falsify a machine learning model. Most frequently, adversarial examples are automatically generated by projected gradient-based methods. Szegedy *et al.* [13] used a box-constrained L-BFGS method to discover the prevalence of adversarial examples in DNN-based classifiers. Goodfellow *et al.* [35] introduced a fast gradient sign method (FGSM), which modifies samples along the steepest descent direction under the $\ell_\infty$-norm constraint. Kurakin *et al.* [36] provided an extension to FGSM, which iteratively generates adversarial examples with a small step size. Madry *et al.* [37] formulated adversarial training as robust optimization [38], where adversarial example generation corresponds to solving the inner maximization problem by projected gradient descent (PGD). The authors recommended a practical trick to construct more transferable adversarial examples: adding a small amount of uniform noise to each pixel of the initial image as a form of dequantization. Moosavi *et al.* [39] defined the adversarial attack as the process to find the minimal perturbation that leads to erroneous model behavior. Close to the present work, Carlini and Wagner [18] considered a Lagrangian relaxation of the $\ell_p$-norm constrained optimization problem to search for adversarial examples. Laidlaw *et al.* [40] switched to a more perceptual FR-IQA model, LPIPS [3], under which adversarial training gives improved robustness to unseen attacks. Here, we consider the Lagrangian formulation in a different context (*i.e.*, NR-IQA), for a different purpose (*i.e.*, encouraging generating samples below JNDs), and under different FR-IQA models (*i.e.*, the Chebyshev distance, SSIM, LPIPS, and DISTS).

We conclude this section by putting adversarial attacks in a broader context of "analysis by synthesis", which is a core idea in the pattern theory by [24]. Analysis by synthesis suggests to test a machine learning model in generative rather than discriminative ways, which is well demonstrated in the field of texture modeling [41]. Adversarial attacks can be seen as a form of analysis by synthesis, where the machine learning model is tested on the automatically generated worst-case data samples, which are less likely to be included in the fixed and excessively reused test sets [42]. In the context of computational vision and neuroscience, the MAD competition [19], the eigen-distortion analysis [20], and the controversial stimuli synthesis [43] all fall into the category of analysis by synthesis. As one of our main contributions, we will make insightful connections of the proposed perceptual attack to these methods.

# 3 Perceptual Attacks on NR-IQA Models

We first formulate the perceptual attack in NR-IQA, and then put it in proper context by making connections to previously related techniques from different engineering fields.

### 3.1 Problem Formulation

An adversarial attack aims to modify a benign data sample $x_0$ to $x^\star$ such that the prediction of a machine learning model $f_w(\cdot)$, parameterized by a vector $w$, undesirably deviates from the ground-truth label $c = f(x_0)$, where $f(\cdot)$ denotes the underlying true hypothesis. This problem can be generally formulated as

$$x^\star = \arg\max_x L(f_w(x), f(x_0)), \text{ s.t. } D(x, x_0) \leq T, \tag{1}$$

where $L(\cdot, \cdot)$ denotes the loss function for a particular machine learning task, $D(\cdot, \cdot)$ is a signal fidelity metric to define the feasible set of perturbations, and $T$ constrains the maximum magnitude of all possible perturbations. In image classification, one may adopt the cross-entropy function as $L(\cdot, \cdot)$ (or simply the negative logit of the ground-truth category for untargeted attacks and the logit difference of the specified and the ground-truth categories for targeted attacks). $\ell_p$-norm induced metrics are frequently used to implement $D(\cdot, \cdot)$. We refer readers to [44, 45, 46, 47, 48, 49] for instantiations of Eq. (1) for attacking other vision tasks.

It is tempting to reuse Eq. (1) to instantiate the adversarial attack on NR-IQA models, where $f_w : \mathbb{R}^M \mapsto \mathbb{R}$ takes an image $x \in \mathbb{R}^M$ as input, and computes a real number as the quality estimate. Without loss of generality, we assume a larger $f_w(x)$ indicates higher predicted quality. The true perceptual quality, $f(x)$, referred to as the mean opinion score (MOS) of $x$, can be collected via a standard psychophysical experiment. As a regression task, we may specify $L(\cdot, \cdot)$ as some discrepancy function between the quality prediction of the perturbed image, $f_w(x)$, and the MOS of the initial image $f(x_0)$, *e.g.*, the mean squared error (MSE), $(f_w(x) - f(x_0))^2$. $D(\cdot, \cdot)$ can be implemented by any "perceptual" FR-IQA model.

We point out three caveats of this constrained optimization. First, existing FR-IQA models only provide a rough account for human perception of image quality. For example, the $\alpha$-level set of $D(\cdot, \cdot)$ w.r.t. $x_0$, *i.e.*, $\{x | D(x, x_0) = \alpha\}$, may contain two images of drastically different quality. Computational models that are specifically designed for measuring JNDs [16, 50, 51, 52, 53, 54, 55], are only tested under simple perturbations (*e.g.*, noise shaping and compression), and may have their own adversarial examples. As a consequence, the feasible set $D(x, x_0) \leq T$ is little likely to have the quality-preserving property. Second, working with an imperfect FR-IQA model requires setting a content-dependent threshold $T(x_0)$ to ensure a quality-preserving feasible set. However, this is a highly non-trivial task, and may result in a trivial solution (*e.g.*, $T(x_0) \to 0$ due to the inaccuracy of $D(\cdot, \cdot)$). Third, solving the constrained optimization problem using PGD (even with careful step size scheduling) appears to converge very slowly (due to the high nonlinearity and nonconvexity of $D(\cdot, \cdot)$). In addition, PGD tends to make the constraint tight with equality and thus is less likely to promote images that are below JNDs. All these motivate us to consider a Lagrangian relaxation of Problem (1):

$$x^\star = \arg\max_x - D(x, x_0) + \lambda(f_w(x) - f(x_0))^2, \tag{2}$$

where $\lambda \geq 0$ is the Lagrange multiplier. An immediate advantage of Eq. (2) is that it allows simultaneous maximization of the discrepancy term measured by the MSE and minimization of the FR-IQA term as the fidelity constraint, which encourages perturbed images to be below JNDs. An alternative way of viewing Eq. (2) is that we perturb $x_0$ in the quality increase and decrease directions specified by the NR-IQA model $f_w$, respectively, under the constraint of $D(\cdot, \cdot)$, and we choose to optimize along the direction that leads to the best discrepancy-fidelity trade-off. Empirically, we find that, if $x_0$ is of poor quality, it is easier to expose counterexamples in the direction of quality improvement, and vice versa. Meanwhile, it is noteworthy that our discrepancy term relies on the MOS of the initial image, $f(x_0)$, which permits naturally occurring failure examples. This is the case when $f_w$ makes a poor quality prediction of $x_0$, leading to large discrepancy and high fidelity. Nevertheless, $f(x_0)$ can be replaced by $f_w(x_0)$ if they are close.

### 3.2 Perceptually Imperceptible Counterexample Generation

We describe a variant of the steepest ascent method to solve Problem (2). Given the initial image $x_0$, we first add to each pixel some noise sampled uniformly from a discrete set $\{-1/255, 0, 1/255\}$ to trigger the optimization [37]. We then compute the steepest ascent direction $g$ of the objective w.r.t. $x$, and take a step of $\gamma$ along the direction of $g$. The pixel values of the intermediate image

---

**Algorithm 1** Perceptually Imperceptible Counterexample Generation

---

**Require:** An NR-IQA model $f_w(\cdot)$, an FR-IQA model $D(\cdot, \cdot)$ as the fidelity measure, a set of $K$ hyperparameter values $\{\lambda_i\}_{i=1}^K$, and the step size $\gamma$
**Input:** An initial image $x_0$
**Output:** A perceptually imperceptible counterexample $x^\star$

1: **for** $i = 1 \to K$ **do**
2:     $x \leftarrow x_0 + \epsilon$, where $\epsilon$ is randomly sampled from $\{-1/255, 0, 1/255\}$
3:     **while** the maximum iteration is not reached **do**
4:         $\lambda \leftarrow \lambda_i$
5:         Compute the objective value $J \leftarrow -D(x, x_0) + \lambda(f_w(x) - f(x_0))^2$
6:         Compute the steepest ascent direction $g \leftarrow \arg\max_v\{\nabla_x J^T v \mid \|v\|_p = 1\}$
7:         Update the image $x \leftarrow x + \gamma \cdot g$
8:         Clamp $x$ into $[0, 1]$
9:     $y_i \leftarrow x$
10:     Quantize $y_i$ to a 24-bit color image
11: Identify the perceptually imperceptible counterexample $x^\star$ from $\{y_i\}_{i=1}^K$ in a psychophysical experiment

---

are clamped into the valid range, *i.e.*, $[0, 1]$, whenever necessary. After the maximum number of iterations is reached, we quantize it to obtain a 24-bit color image as the output.

By varying the Lagrange multiplier $\lambda$, we are able to generate a sequence of images with different distortion visibility. We next design a psychophysical experiment to identify the perceptually imperceptible counterexample, giving careful treatment to two subtleties. First, the distortion visibility of the generated image generally reduces with the decrease of $D(x^\star, x_0)$ and the decrease of $\lambda$ (see Eq. (2)), but there are exceptions due to the high dimensionality and non-convexity of the optimization problem. Therefore, standard psychophysical staircase methods [56] that assume monotonicity w.r.t. the stimulus intensity may not be straightforwardly applied. Second, ideally, we would like to measure the just-noticeable distortion[2], but practically, it is difficult to collect such human thresholds in a reliable way unless excessive instructions are given. This is especially true when $x_0$ has already been severely distorted (*e.g.*, blurred or compressed), and it is no easy task to determine whether the added perturbation further degrades the image quality.

We thus choose the yes-no task for screening perturbed images below JNDs [57]. Specifically, human participants undergo a series of trials, each consisting of a pair of the perturbed image (corresponding to a certain $\lambda$) and the initial image. For each trial, they must judge whether the two images look identical. A perturbed image is said to be below the JND if participants fail to distinguish it from the initial image 75% of the time. *We identify the perceptually imperceptible counterexample as the perturbed image that 1) is below the JND and 2) causes the largest change in quality prediction.* More details about the psychophysical experiment can be found in Sec. 4.1, and the procedure of counterexample generation of NR-IQA models is summarized in Algorithm 1.

### 3.3 Connections to Previous Work

In this subsection, we make insightful connections to several beautiful ideas in signal processing, computational vision, and machine learning.

**Carlini-Wagner attack** [18] has the closest relationship to the proposed perceptual attack on NR-IQA models with three differences. First, Carlini and Wagner chose to work with the Lagrangian formulation mainly from the perspective of *computational convenience*, while we arrive at the same formulation from a different *perceptual* perspective: no computational models exist to reliably compute JNDs. Second, for classification, there are $C - 1$ directions for the Carlini-Wagner attack to look for adversarial examples, where $C$ is the number of categories. For NR-IQA as regression, there are two such directions. Third, the Carlini-Wagner attack suffices to employ the binary search to find the desired Lagrange multiplier due to the label-preserving property of the majority perturbations in image classification. In contrast, the proposed perceptual attack may have to rely on carefully

---

[2]In some situations, the visible differences may not be perceived as distortions by the human visual system (*e.g.*, image enhancement).

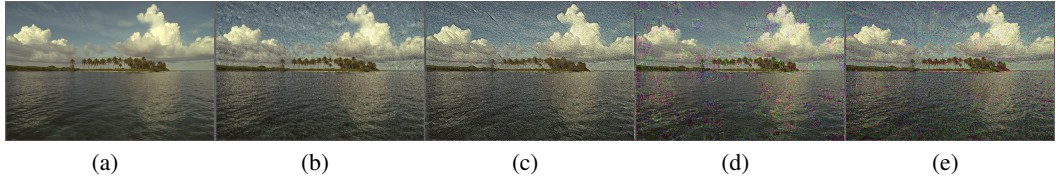

|   (a)   |   (b)   |   (c)   |   (d)   |   (e)   |

Figure 2: "Enhanced" versions from the initial image **(a)** by maximizing BRISQUE **(b)**, CORNIA **(c)**, Ma19 **(d)**, and UNIQUE **(e)**, respectively.

designed psychophysical experiments to determine the value of $\lambda$ in NR-IQA, which is common practice in computational vision.

**MAP estimation** [58, 59, 60, 61] seeks to estimate a latent image $x^\star$ given a corrupted observation $x_0$ by maximizing the posterior probability $p(x|x_0)$:

$$x^\star = \arg \max_x p(x|x_0) = \arg \max_x \log p(x_0|x) + \lambda \log p(x), \qquad (3)$$

where $\log p(x_0|x)$ is the data fidelity term implemented by an FR-IQA model (*e.g.*, the MSE) and $\log p(x)$ is the image prior term to measure the naturalness of the image. A better prior leads to improved estimation performance. The added hyperparameter $\lambda$ is often manually tuned, where $\lambda = 1$ makes Eq. (3) hold. It is widely acknowledged that an ideal NR-IQA model must rely solely on knowledge of the appearance of natural undistorted images. This suggests that the NR-IQA method should embody an image prior model, and perhaps even that the quality predictions should be monotonically related to probability densities [22]. With such conceptual equivalence between NR-IQA methods and natural image priors, we see that the proposed perceptual attack differs only from MAP estimation in the choice of $\lambda$. Specifically, our attack sets a $\lambda$ to produce the image below the JND relative to $x_0$ while causing the most significant change of $f_w(\cdot)$. In contrast, MAP estimation tests $p(x)$ using a $\lambda$ that leads to the best estimation performance. More importantly, we shall come to a conclusion: in order for NR-IQA models to work in MAP estimation, they must first survive the proposed perceptual attack as an easier task. Taking a step further, we may set $\lambda = +\infty$, which corresponds to the maximization of $f_w(\cdot)$ alone as a way of performing image enhancement. Not surprisingly, all state-of-the-art NR-IQA models fail this task, generating images with annoying distortions but higher predicted quality scores (see Fig. 2).

**MAD competition** [19] is an efficient methodology for comparing computational methods of perceptually discriminable quantities (*e.g.*, image quality). In the context of comparing two IQA models, $f_1(\cdot)$ and $f_2(\cdot)$, MAD automatically synthesizes an image pair $(x^\star, y^\star)$ that are likely to falsify at least one IQA model in competition by solving

$$(x^\star, y^\star) = \arg \max_{x,y} \ f_1(x) - f_1(y), \ \text{s.t.} \ f_2(x) = f_2(y) = f_2(x_0), \qquad (4)$$

where $x_0$ is the initial image and $f_2(x_0)$ specifies a quality level. By doing so, we obtain a pair of counterexamples, on which $f_1(\cdot)$ and $f_2(\cdot)$ hold contradictory opinions: $f_1(\cdot)$ predicts $x^\star$ to have much better quality than $y^\star$, while $f_2(\cdot)$ treats them as images of identical quality. Similar as the proposed perceptual attack, psychophysical testing on $(x^\star, y^\star)$ is required to declare which model is the winner. In contrast to our attack that is constrained in the set of images that look identical to $x_0$ according to human perception (approximated by an FR-IQA model), MAD operates at the level set of another IQA model $f_2(\cdot)$ that contains images of the same predicted quality as $x_0$. Moreover, MAD generates the best-quality and worst-quality images in terms of $f_1(\cdot)$ to compare the relative performance of two IQA models, while we only test the perceptual robustness of a single NR-IQA model.

**Eigen-distortion analysis** [20] is a computational method for comparing perceptual image representations, $f : \mathbb{R}^M \mapsto \mathbb{R}^N$. It computes the eigenvectors of the Fisher information matrix[3] with the largest and smallest eigenvalues to account for the model-predicted most- and least-noticeable distortion directions, respectively. A psychophysical experiment based on a standard staircase method

---

[3]Assuming the additive white Gaussian noise in the response space, we can compute the Fisher information matrix by $\frac{\partial f}{\partial x}^T \frac{\partial f}{\partial x}$, where $\frac{\partial f}{\partial x}$ is the Jacobian matrix.

is necessary to measure the ratio of the visibility thresholds induced by the two directions. When $N = 1$ (*i.e.*, $f(\cdot)$ can be an NR-IQA method), the most-noticeable direction reduces to the gradient of $f$ w.r.t. the input image $x$, $\partial f/\partial x$. Correspondingly, the least-noticeable direction (with an eigenvalue of zero) can be any $M$-dim unit vector living in the $M-1$-dim subspace orthogonal to $\partial f/\partial x$. Taking consecutive small steps along least-noticeable directions creates a predicted imperceptible perturbation, which is subject to human verification. In this sense, the proposed perceptual attack aims to tackle a "dual" problem, generating a true imperceptible perturbation to maximize the prediction discrepancy.

## 4 Experiments

In this section, we first set up the experiments, including descriptions of NR-IQA and FR-IQA models, details of the psychophysical experiment, and new evaluation metrics for measuring the perceptual robustness of NR-IQA models. We then present the quantitative and qualitative results accompanied by an in-depth analysis.

### 4.1 Experimental Setups

**Choice of NR-IQA Models**. We test four NR-IQA models that are believed to be representative in the field: **BRISQUE** [17], **CORNIA** [21], **Ma19** [22], and **UNIQUE** [6]. BRISQUE is a knowledge-driven model, extracting NSS from mean-subtracted contrast-normalized pixel values. CORNIA [21] relies on unsupervised learning of a visual codebook from image patches, followed by soft-assignment coding and max pooling, to obtain image representations. As feature engineering is not involved, CORNIA can be seen as a shallow learning-based method. Ma19 [22] is a four-layer DNN with generalized divisive normalization [62] as nonlinear activation, which is trained from a large number of corrupted image pairs without reliance on MOSs. UNIQUE [6] trains a variant of ResNet-34 [1] on multiple IQA datasets to handle both synthetic and realistic camera distortions. We use the training codes provided by the original authors to re-train BRISQUE and CORNIA on LIVE [7], Ma19 [22] on our own collected dataset[4], and UNIQUE on six human-rated IQA databases [7, 63, 64, 9, 5, 65]. There is no overlap between all training sets and the initial images used to generate counterexamples.

As suggested by the Video Quality Experts Group [66], a four-parameter logistic function can be adopted to compensate for the prediction nonlinearity, making different NR-IQA more comparable:

$$q \circ f_w(x) = \frac{\beta_1 - \beta_2}{1 + \exp^{-\frac{f_w(x) - \beta_3}{|\beta_4|}}} + \beta_2, \tag{5}$$

where $\circ$ indicates the function composition operation. $\beta_1$ to $\beta_4$ are fitting parameters, where $\beta_1$ and $\beta_2$ determine the maximum and minimum mapping values. Empirically, we find that, for different NR-IQA models, the estimated $\beta_1$ and $\beta_2$ can be quite different. Thus, we choose to manually enforce $\beta_1 = 10$ and $\beta_2 = 0$. We consider the learned $q(\cdot)$ as part of the NR-IQA model.

**Choice of FR-IQA Models**. We consider four FR-IQA models to approximate the perceptual distance between the initial and perturbed images: the **Chebyshev** distance, **SSIM**[5] [10], **LPIPS** [3], and **DISTS** [4]. The Chebyshev distance constrains the maximum pixel difference within an $\ell_\infty$-ball. SSIM [10] is arguably the most successful "perceptual" metric that compares luminance, contrast, and structure, separately. LPIPS [3] computes the Euclidean distance between deep representations of two images, which shows reasonable effectiveness in explaining image quality. DISTS [4] is the first FR-IQA method that unifies structure and texture similarity, and is competitive in perceptual optimization of various image processing tasks [67]. We apply the Chebyshev distance to each color channel, and take the maximum distance across three channels. We enable SSIM to be aware of color information by treating the color-to-grayscale conversion as a fixed differentiable front-end, which allows the gradient to be back-propagated to the input color image. Both LPIPS and DISTS are based on variants of VGG-16 [68].

**Details of the Psychophysical Experiment**. We collect twelve images as initializations from the publicly available LIVE IQA database [7] (see Fig. 3), with four basic distortions (*i.e.*, JPEG

---

[4]We use the same pristine-quality images in [22], but generate the training set with only four types of distortions overlapping with LIVE, *i.e.*, white Gaussian noise, Gaussian blur, JPEG compression, and JPEG2000 compression.

[5]In our implementation, we negate SSIM to make it a distance measure.

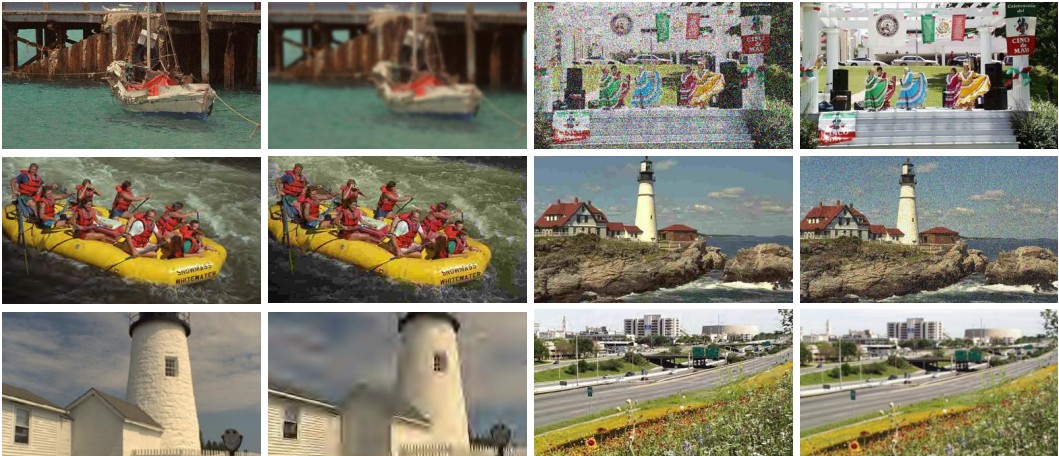

Figure 3: Twelve initial images from the LIVE database [7]. Images are cropped for improved visibility.

compression, JPEG2000 compression, Gaussian blur, and additive white Gaussian noise) in the field of IQA. In addition, we ensure that the selected images cover a broad quality range with discriminable perceptual quality. For each of sixteen combinations of NR-IQA and FR-IQA models, and each of the twelve initial images, we set $\lambda$ to 32 values, and optimize the objective in Eq. (2) to generate 32 perturbed images. For the Chebyshev distance, we use the steepest ascent direction in $\ell_\infty$-norm, and for the remaining three FR-IQA models, we choose the steepest ascent direction in $\ell_2$-norm, namely the gradient direction. We set the step size $\gamma$ to $10^{-3}$ and the maximum number of iterations to 200, respectively. As suggested in the BT. 500 recommendations [69], we carry out the experiments in an indoor office environment with a normal lighting condition (with approximately 200 lux) and without reflecting ceiling walls and floors. The peak luminance of the displayed images is mapped to 200 $\mathrm{cd/m^2}$. We recruit fifteen human subjects (with normal or corrected-to-normal vision) to participate in the psychophysical experiment, viewing the image pairs from a fixed distance of twice the screen height. A training session is performed to familiarize each subject with the study. For each yes-no trial, subjects are shown (for one second each with a half-second gray screen between images, and in randomized order) a perturbed image and the corresponding initial image, and then asked to determine whether the two images look visually different. All image pairs are displayed on a $24''$ LCD monitor at a resolution of $1,920 \times 1,080$. To avoid the fatigue effect, subjects are required to take a break during the experiment. In total, we generate $4 \times 4 \times 12 = 192$ counterexamples to test the robustness of NR-IQA models.

**Evaluation Metrics**. Unlike image classification, it is generally difficult (or even pointless) for a regression task to define the success of an adversarial attack on a per-image basis. As image quality is a *relative* perceptual quantity, we adopt the Spearman rank-order correlation coefficient (SRCC) to measure the prediction monotonicity between model predictions and MOSs of 24 images (12 initial images and 12 corresponding counterexamples). In addition, we define the average ratio of maximum allowable change in quality prediction to actual change over counterexamples in a logarithmic scale:

$$R = \frac{1}{S}\sum_{i=1}^{S}\log\left(\frac{\max\left\{\beta_1 - f_w(x_i), f_w(x_i) - \beta_2\right\}}{|f_w(x_i) - f_w(x_i^\star)|}\right), \tag{6}$$

where $S$ is the number of initial images and $x_i$ denotes the $i$-th one. A larger $R$ means better stability (*i.e.*, robustness), but not necessarily better quality prediction performance as the MOS is not involved in the computation.

### 4.2 Main Results

We summarize the SRCC and average ratio results of NR-IQA models under intra-model attacks (*i.e.*, using counterexamples generated by the proposed attack with different FR-IQA models) and

Table 1: SRCC and average ratio results of NR-IQA models under intra-model attacks

| Metric | SRCC | | | | $R$ defined in Eq. (6) | | | |
|---|---|---|---|---|---|---|---|---|
| FR-IQA | Chebyshev | SSIM | LPIPS | DISTS | Chebyshev | SSIM | LPIPS | DISTS |
| BRISQUE | 0.0959 | 0.2023 | 0.1221 | 0.1308 | 0.4116 | 0.3824 | 0.4412 | 0.7392 |
| CORNIA | 0.1998 | 0.0663 | 0.2092 | 0.0139 | 0.9141 | 0.4181 | 0.3682 | 0.2197 |
| Ma19 | 0.2441 | 0.1447 | 0.0959 | 0.2005 | 0.6188 | 0.3914 | 0.3943 | 0.3828 |
| UNIQUE | -0.0994 | -0.1064 | -0.1813 | -0.1726 | 0.2059 | 0.1539 | 0.0927 | 0.1229 |

Table 2: SRCC and average ratio results of NR-IQA models under inter-model attacks. The number in the parentheses shows the original performance of the NR-IQA model

| | Attack | BRISQUE (0.9231) | | | CORNIA(0.9650) | | |
|---|---|---|---|---|---|---|---|
| | Under Attack | CORNIA | Ma19 | UNIQUE | BRISQUE | Ma19 | UNIQUE |
| | Chebyshev | 0.8230 | 0.6103 | 0.7777 | 0.8596 | 0.9730 | 0.9660 |
| | SSIM | 0.8684 | 0.9091 | 0.9276 | 0.9538 | 0.9790 | 0.9555 |
| | LPIPS | 0.7725 | 0.8806 | 0.8806 | 0.9154 | 0.9451 | 0.9451 |
| SRCC | DISTS | 0.7254 | 0.8806 | 0.8806 | 0.8997 | 0.9660 | 0.9625 |
| | Attack | Ma19 (0.9091) | | | UNIQUE(0.9301) | | |
| | Under Attack | BRISQUE | CORNIA | UNIQUE | BRISQUE | CORNIA | Ma19 |
| | Chebyshev | 0.7725 | 0.8387 | 0.8422 | 0.9381 | 0.9224 | 0.9242 |
| | SSIM | 0.8771 | 0.9102 | 0.9137 | 0.9346 | 0.9137 | 0.9276 |
| | LPIPS | 0.8718 | 0.8387 | 0.8893 | 0.9276 | 0.9346 | 0.9346 |
| | DISTS | 0.8387 | 0.8422 | 0.8858 | 0.9276 | 0.8945 | 0.9259 |
| | Attack | BRISQUE ($+\infty$) | | | CORNIA ($+\infty$) | | |
| | Under Attack | CORNIA | Ma19 | UNIQUE | BRISQUE | Ma19 | UNIQUE |
| | Chebyshev | 2.1064 | 2.7651 | 2.4192 | 3.2428 | 4.3930 | 4.2005 |
| | SSIM | 2.4097 | 3.1472 | 3.2371 | 3.3378 | 4.1283 | 4.2050 |
| | LPIPS | 1.7400 | 3.1070 | 3.0490 | 3.5316 | 4.2963 | 4.6841 |
| $R$ defined in Eq. (6) | DISTS | 1.8017 | 3.1368 | 2.9193 | 3.8798 | 3.8258 | 4.0432 |
| | Attack | Ma19 ($+\infty$) | | | UNIQUE ($+\infty$) | | |
| | Under Attack | BRISQUE | CORNIA | UNIQUE | BRISQUE | CORNIA | Ma19 |
| | Chebyshev | 2.4957 | 3.0544 | 3.1594 | 4.0762 | 3.6509 | 3.9139 |
| | SSIM | 3.6393 | 4.3902 | 4.4728 | 4.6850 | 2.9966 | 4.2564 |
| | LPIPS | 2.9033 | 3.0309 | 3.5103 | 4.4247 | 3.2835 | 4.9144 |
| | DISTS | 3.6089 | 3.4737 | 3.5533 | 4.8580 | 2.6109 | 4.1454 |

inter-model attacks (*i.e.*, using counterexamples originally spotted to falsify another NR-IQA model) in Tables 1 and 2, respectively. The results without perceptual attacks are also presented as reference. From the tables, we have a number of insightful observations.

First, the intra-attacks are capable of falsifying the corresponding NR-IQA models, no matter which FR-IQA model is adopted. This is evidenced by a catastrophic SRCC drop relative to the original performance. We then conclude that none of the evaluated NR-IQA design philosophies (*i.e.*, NSS-based, codebook-based, and DNN-based approaches), are inherently perceptually robust. Second, compared with intra-model attacks, NR-IQA methods appear much more robust to inter-model attacks. This manifests the poor transferability of perceptually imperceptible counterexamples, which fail to craft black-box attacks in NR-IQA. On the positive side, the generated examples by the proposed perceptual attack are informative to reveal distinct design flows of different NR-IQA models, and may point out promising ways to improve a model or to combine the best aspects of multiple models. Third, when switching to the average ratio ($R$ defined in Eq. (6)), we find that UNIQUE is the least stable model consistently across all FR-IQA methods, suggesting that the overparameterization of UNIQUE creates an abundant space for overfitting the current IQA datasets. Although the observations have been obtained from the expensive psychophysical experiment on only twelve images, we empirically find these to be consistent across a wide range of images with different content and distortion complexities.

To compare the perturbations generated for different NR-IQA models in a more intuitive way, we visualize the absolute residual images (*i.e.*, ($|x_0 - x^\star|$) in Fig. 4, where DISTS is used as the image fidelity measure in Eq. (2). The primary observation is that the difference in perturbations provides useful clues on how they extract quality-aware features. Specifically, perturbations for BRISQUE [17]

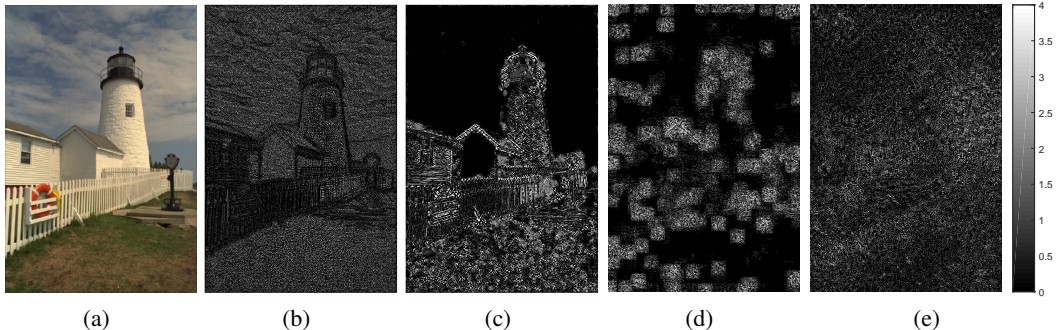

Figure 4: Perturbations added to the initial image **(a)** by attacking BRISQUE **(b)**, CORNIA **(c)**, Ma19 **(d)**, and UNIQUE **(e)**, respectively.

mainly emerge in smooth and textured regions, where manipulation of individual and product of locally normalized luminances is much easier, as a way of cheating the built NSS model. The learned codebook in CORNIA [21] contains many Dirac delta functions of different locations and edge filters of different orientations, for which the selected texture and edge patches from a test image give the maximum response and thus are used for quality computation. This may explain the perturbations in Fig. 4 (c) are concentrated along the edges and in strong textures. The blocking perturbations in Fig. 4 (d) appear primarily on the objects (*e.g.*, the lighthouse and the fence) and along strong edges (*e.g.*, the image borders due to zero padding in DNNs). We believe this arises from the spatial pyramid pooling [70] layer used in Ma19 [22]. When the pyramid pooling is replaced with global average pooling, the perturbations by Ma19 resemble those in Fig. 4 (e) by UNIQUE. In addition, compare to BRISQUE and CORNIA that only accept grayscale images, the perturbations by Ma19 and UNIQUE occur in all color channels (not shown). Finally, nearly all pixel perturbations are less than $4/255$, justifying the effectiveness of our psychophysical experiment to identify counterexamples that are below JNDs.

## 5  Conclusion

We have described a perceptual attack on NR-IQA models, based on which their perceptual robustness has been thoroughly evaluated. The proposed attack has close connections to previous techniques. We found that neither the conventional knowledge-driven NR-IQA models nor the modern DNN-based methods are inherently robust to perceptually imperceptible perturbations. Moreover, the generated counterexamples by one NR-IQA model do not transfer in an efficient way to falsify other models, which contain valuable information to expose the design flows of respective models. We believe with the exploration of perceptual attacks of NR-IQA models, NR-IQA research may enter a new and explosive development stage.

In the future, we plan to develop perceptually robust NR-IQA methods. From the model construction perspective, we will identify and combine robust building blocks of different NR-IQA methods. From the optimization perspective, we may directly employ adversarial training as suggested by [37]. Meanwhile, we may draw inspiration from the connections between the proposed perceptual attack and previous analysis by synthesis techniques, and develop robust regularizers [71] to facilitate training robust NR-IQA methods.

## Acknowledgments and Disclosure of Funding

The authors would like to thank Chengguang Zhu for coordinating the psychophysical experiment and all subjects for participation. This work was supported in part by the Hong Kong RGC ECS Grant 21213821, the National Natural Science Foundation of China under Grants 62071407 and 61901262, and Shanghai Municipal Science and Technology Major Project (2021SHZDZX0102).

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
