# OpenReview forum: "Perceptual Attacks of No-Reference Image Quality Models with Human-in-the-Loop"
_NeurIPS.cc/2022/Conference — NeurIPS 2022 Accept_

### Official Review · Reviewer_4XSx · 2022-07-07

**Rating:** 5
**Confidence:** 3
**Soundness:** 2 fair
**Presentation:** 2 fair
**Contribution:** 2 fair

**Summary:**

This paper aims to investigate the perceptual robustness of NR-IQA models. It proposes a Lagrangian relaxation of the existing adversarial attack formula. Then it gives the algorithm for perceptually imperceptible counterexample generation. The experimental results show some findings.

**Questions:**

1. Why this problem is important?

2. Are there any suggestions that can be concluded from the findings to defend against the adversarial attack?

3. How do you make sure that the perturbation is below the just-noticeable difference? The examples computed by MAP estimation shown in Fig. 2 are clearly different from the original image. The examples of the proposed method are not given in the paper. So it is hard to judge whether the perturbation is noticeable or not.

4. What is the relationship between the four values in Italics and the value in parentheses for each group?


**Limitations:**

The limitations are not discussed in the paper.

**Strengths And Weaknesses:**

Strengths:
The problem is novel.
The problem and its relationship with adversarial attacks in classification are well described. The Lagrangian relaxation and the algorithm for perceptually imperceptible counterexample generation are easy to follow.


Weaknesses:
The motivations to investigate the perceptual robustness of NR-IQA models should be given. Why this problem is important?
The findings are a bit general. What are the insights for the application of NR-IQA metrics? Are there any suggestions that can be concluded from the findings to defend against the adversarial attack?

---

> ### Author Response · Authors · 2022-07-28
> **Responses to Reviewer 4XSx: (1)**
>
> Thanks for the valuable comments, pointing out current portions that need to be further polished and future directions that are worth exploring. Point-for-point comments are listed below.
>
> **Regarding the importance of IQA**: Image quality assessment, full-reference or no-reference, is a fundamental and long-standing problem in image processing, computer vision, and cognitive science. IQA has a long history with many FR-IQA and NR-IQA methods proposed, but, so far, there still lack reliable and robust NR-IQA models to bring revolutionary improvements to corresponding applications, e.g., image restoration, colorization, deblurring, deraining, etc., in both academic and industry. With the exploration of perceptual attacks of NR-IQA models where reference images cannot be accessed, NR-IQA research may enter a new and explosive development stage, and there are good opportunities to design robust NR-IQA models to be used in various real-world quality applications. We are sure that NR-IQA is still a hot topic worthy of study.
>
> **Regarding defending against the perceptual attack**:  Thanks for the valuable comment, which is indeed what we are currently exploring. One of the key observations is that the counterexamples generated from one NR-IQA model are not transferrable to falsify other models. As suggested by the Reviewer qeax, we may directly leverage model ensemble tricks to mitigate the vulnerability of perceptual quality metrics, but it turns out not a viable solution.
> We have also conducted some preliminary experiments by following the common practice of using the method of Madry et al. for adversarially training the UNIQUE model. Counterexamples during training are generated under an $\ell_{\infty}$-constraint with a radius of $\frac{4}{255}$, and the number of attacking iterations and the step size are set to $5$ and $\frac{2}{255}$, respectively. After training, we test the adversarially trained model using the proposed perceptual attack, and conduct an informal psychophysical experiment to verify the perceptual imperceptibility of the found adversarial examples (by two of the authors). As shown in the table below, vanilla adversarial training significantly improves the robustness of the UNIQUE model. As suggested by the reviewer, in the future, we will delve deep into the design and training of adversarially robust NR-IQA models with improved effectiveness and efficiency.
>
> Table R4 SRCC and average ratio results of the UNIQUE model after the standard and adversarial training.
> | Standard Training |        | Adversarial Training |        |
> |:-----------------:|:------:|:--------------------:|:------:|
> |        SRCC       |    R   |         SRCC         |    R   |
> |      -0.0994      | 0.2059 |        0.6382        | 1.6395 |
>
> **Regarding ensuring the perceptual imperceptibility of the perturbation**: Thanks for the comment. To make sure that the perturbation is below the JND, we have conducted a well-controlled psychophysical experiment, where human participants undergo a series of trials, each consisting of a pair of the perturbed image (corresponding to a certain $\lambda$) and the initial image. For each trial, they must judge whether the two images look identical. A perturbed image is said to be below the JND if participants fail to distinguish it from the initial image 75% of the time.
>
> The examples shown in Fig. 2 are not the results of the proposed perceptual attacks. They are the results of the MAP estimation, where $\lambda = +\infty$. In such cases, NR-IQA models predict the corresponding images to be of high quality, but the human eye can clearly see the distortions in the images, leading to poor perceived quality. From this ``naive’’ experiment, we conclude in the original manuscript that in order for NR-IQA models to work in MAP estimation, they must first survive the proposed perceptual attack as an easier task. In the Appendix, we have shown some counterexamples generated by the proposed perceptual attack with amplified perturbations (otherwise they are invisible). As suggested by the reviewer, we will include counterexamples in the appendix without perturbation amplification.

---

> > ### Author Response · Authors · 2022-07-28
> > **Responses to Reviewer 4XSx: (2)**
> >
> > **Regarding Table 1**: Thanks for the comments. The four values in Italics are the results of intra-model attacks. For example, in the top-left section of the current Table 1, the four values in Italics are obtained by testing BRISQUE on the counterexamples generated by attacking BRISQUE itself. The value in the parentheses indicates the original performance of an NR-IQA model, e.g., BRISQUE attains an SRCC of $0.9231$ on initial images without any attack. As also suggested by the Reviewer 1irv, we have divided the current Table 1 into two tables, one for intra-model attacks (i.e., counterexamples spotted to falsify the corresponding NR-IQA model by solving Problem (2)) and the other for inter-model attacks (i.e., counterexamples spotted to falsify other irrelevant NR-IQA models), as shown below. We have also rephrased the caption to make it much clearer.
> >
> > Table R1 SRCC and average ratio results of NR-IQA models under intra-model attacks.
> > |  Metric |    SRCC   |   |   |   |     R     |       |       |       |
> > |:-------:|:---------:|:-------:|:-------:|:-------:|:---------:|:------:|:------:|:------:|
> > |  NR-IQA | Chebyshev |   SSIM  |  LPIPS  |  DISTS  | Chebyshev |  SSIM  |  LPIPS |  DISTS |
> > | BRISQUE |   0.0959  |  0.2023 |  0.1221 |  0.1308 |   0.4116  | 0.3824 | 0.4412 | 0.7392 |
> > |  CORNIA |   0.1998  |  0.0663 |  0.2092 |  0.0139 |   0.9141  | 0.4181 | 0.3682 | 0.2197 |
> > |   Ma19  |   0.2441  |  0.1447 |  0.0959 |  0.2005 |   0.6188  | 0.3914 | 0.3943 | 0.3828 |
> > |  UNIQUE |  -0.0994  | -0.1064 | -0.1813 | -0.1726 |   0.2059  | 0.1539 | 0.0927 | 0.1229 |
> >
> > Table R2 SRCC and average ratio results of NR-IQA models under inter-model attacks. The number in the parentheses shows the original performance of the NR-IQA model.
> > |   Metric  |         SRCC        |        |        |        SRCC        |        |        |
> > |:---------:|:-------------------:|:------:|:------:|:------------------:|:------:|:------:|
> > |   Attack  |   BRISQUE (0.9231)  |        |        |   CORNIA(0.9650)   |        |        |
> > |   Under Attack |        CORNIA       |  Ma19  | UNIQUE |       BRISQUE      |  Ma19  | UNIQUE |
> > | Chebyshev |        0.8230       | 0.6103 | 0.7777 |       0.8596       | 0.9730 | 0.9660 |
> > |    SSIM   |        0.8684       | 0.9091 | 0.9276 |       0.9538       | 0.9790 | 0.9555 |
> > |   LPIPS   |        0.7725       | 0.8806 | 0.8806 |       0.9154       | 0.9451 | 0.9451 |
> > |   DISTS   |        0.7254       | 0.8806 | 0.8806 |       0.8997       | 0.9660 | 0.9625 |
> > |   Metric  |         SRCC        |        |        |        SRCC        |        |        |
> > |   Attack  |    Ma19 (0.9091)    |        |        |   UNIQUE(0.9301)   |        |        |
> > |   Under Attack  |       BRISQUE       | CORNIA | UNIQUE |       BRISQUE      | CORNIA |  Ma19  |
> > | Chebyshev |        0.7725       | 0.8387 | 0.8422 |       0.9381       | 0.9224 | 0.9242 |
> > |    SSIM   |        0.8771       | 0.9102 | 0.9137 |       0.9346       | 0.9137 | 0.9276 |
> > |   LPIPS   |        0.8718       | 0.8387 | 0.8893 |       0.9276       | 0.9346 | 0.9346 |
> > |   DISTS   |        0.8387       | 0.8422 | 0.8858 |       0.9276       | 0.8945 | 0.9259 |
> > |   Metric  |          R          |        |        |          R         |        |        |
> > |   Attack  | BRISQUE ($+\infty$) |        |        | CORNIA ($+\infty$) |        |        |
> > |   Under Attack  |        CORNIA       |  Ma19  | UNIQUE |       BRISQUE      |  Ma19  | UNIQUE |
> > | Chebyshev |        2.1064       | 2.7651 | 2.4192 |       3.2428       | 4.3930 | 4.2005 |
> > |    SSIM   |        2.4097       | 3.1472 | 3.2371 |       3.3378       | 4.1283 | 4.2050 |
> > |   LPIPS   |        1.7400       | 3.1070 | 3.0490 |       3.5316       | 4.2963 | 4.6841 |
> > |   DISTS   |        1.8017       | 3.1368 | 2.9193 |       3.8798       | 3.8258 | 4.0432 |
> > |   Metric  |          R          |        |        |          R         |        |        |
> > |   Attack  |   Ma19 ($+\infty$)  |        |        | UNIQUE ($+\infty$) |        |        |
> > |   Under Attack  |       BRISQUE       | CORNIA | UNIQUE |       BRISQUE      | CORNIA |  Ma19  |
> > | Chebyshev |        2.4957       | 3.0544 | 3.1594 |       4.0762       | 3.6509 | 3.9139 |
> > |    SSIM   |        3.6393       | 4.3902 | 4.4728 |       4.6850       | 2.9966 | 4.2564 |
> > |   LPIPS   |        2.9033       | 3.0309 | 3.5103 |       4.4247       | 3.2835 | 4.9144 |
> > |   DISTS   |        3.6089       | 3.4737 | 3.5533 |       4.8580       | 2.6109 | 4.1454 |

---

### Official Review · Reviewer_Hmqo · 2022-07-10

**Rating:** 5
**Confidence:** 3
**Soundness:** 3 good
**Presentation:** 2 fair
**Contribution:** 3 good

**Summary:**

This submission proposes a Lagrangian formulation to study the perceptual attack of no-reference image quality. By using the Lagrangian relaxation between the prediction error of NR-IQA and the perceptual constraints with FR-IQA, the perceptually imperceptible counterexamples are generated. With the psychophysical experiments, the analysis of the perceptual attack of NR-IQA is conducted.

**Questions:**

1）What is the basis for choosing these 6 images and their corresponding distortions？

**Ethics Review Area:**

["Inadequate Data and Algorithm Evaluation", "Privacy and Security (e.g., consent)"]

**Strengths And Weaknesses:**

Strengths:
1)	perceptual attack of NR-IQA is an interesting problem, which would benefit to the image quality assessment community.
2)	The visualization of the perturbations to the different NR-IQA methods shows the different characteristics, which is also interesting.

Weaknesses:
1)	It states the connections with the existing methods, while it is better to validate the connections by experiments.
2)	It lacks comparisons, making it hard to judge its advancement.
3)	It is better to provide the statistic for the relationship between the Lagrange multiplier and the perceptual quality.
4)	The original performance of R in Table 1 is positive infinity, and the following values are not informative. Equation 6 can be modified by adding a regular term to the denominator to make the original performance a positive real number.
5)	SSIM is a higher the better metric, and it should be stated in the text whether (1-SSIM) or -SSIM is used as the D

---

> ### Author Response · Authors · 2022-07-28
> **Responses to Reviewer Hmqo: (1)**
>
> Thanks for the valuable comments, pointing out current portions that need to be further polished and future directions that are worth exploring. Point-for-point comments are listed below.
>
> **Regarding validating the connections to existing methods by experiments**: Thanks for the comment. In the original manuscript, we have connected the proposed perceptual attack to several beautiful ideas in signal processing, computational vision, and machine learning, as they all fall into the category of ``analysis by synthesis’’, as well recognized by the Reviewer 1irv. However, it is difficult (if not impossible) to design computational experiments to validate the connections as they target for different tasks. We believe those conceptual connections are equally important to contextualize the proposed perceptual attack, and to inspire future attacks based on analysis by synthesis. For example, there will be a chance for an NR-IQA model to work in MAP estimation if it can survive the proposed perceptual attack as an easier task.
>
> **Regarding lacking comparison for advancement justification**: Thanks for the comment. As the first work of studying perceptual attack for NR-IQA, we are only able to compare different NR-IQA models as targets and different FR-IQA as perceptual constraints in the manuscript. This is well recognized by Reviewers 1irv and qeax, as one of our key contributions. The most related work is the Carlini-Wagner attack on image classification, which adopts Euclidean distance (i.e., the mean squared error (MSE) in the pixel domain) as the perceptual constraint. As suggested by Reviewer 1irv, we further compare DISTS against MSE as the perceptual constraint in the below table, from which we find that a more perceptual FR-IQA model generally delivers more effective attacks, spotting counterexamples that are below JNDs yet more powerful.
>
> Table R3 SRCC and average ratio results using MSE and DISTS as the perceptual constraints.
> | FR-IQA | Metric | BRISQUE | CORNIA |  Ma19  |  UNIQUE |
> |:------:|:------:|:-------:|:------:|:------:|:-------:|
> |   MSE  |  SRCC  |  0.2128 | 0.2934 | 0.2568 |  0.1779 |
> |        |    R   |  0.4616 | 1.7028 | 0.7273 |  0.3687 |
> |  DISTS |  SRCC  |  0.1308 | 0.0139 | 0.2005 | -0.1726 |
> |        |    R   |  0.7392 | 0.2197 | 0.3828 |  0.1229 |
>
>
> **Regarding statistical relationship between the Lagrange multiplier and the perceptual quality**: Thanks for the comment. Many of the signal fidelity terms (e.g., SSIM, LPIPS, DISTS) and all the discrepancy terms (e.g., BRISQUE, CORNIA, Ma19, and UNIQUE) weighted by $\lambda$ in Eq. (2) are highly nonlinear and nonconvex with respect to the high-dimensional input image. It is thus a highly nontrivial task to derive a rigorous statistical relationship between Lagrange multiplier and perceptual quality (even approximated by the adopted signal fidelity term). Nevertheless, we empirically observe that in most cases, increasing $\lambda$ generally leads to the decrease of the signal fidelity term $D(\cdot)$.
>
> **Regarding adding a regularization term in Eq. (6)**: Thanks for the comment. The definition of $R$ is primarily inspired by the Peak Signal-to-Noise Ratio (PSNR) widely used in signal and image processing, which can take on positive infinity. By definition, an NR-IQA model with perfect perceptual robustness will make exactly the same prediction for an initial image $x$ and the corresponding counterexample $x^{\star}$, correct or not. In that case, the denominator of Eq. (6) $\| f_{w}(x) – f_{w}(x^{\star}) \|$ is zero, and the value of $R$ would be positive infinity. We respectfully disagree with the Reviewer that Eq. (6) is better modified by adding a regularization term (e.g., a small positive constant) to the denominator. This is because if we do so, the original performance will depend on the initial prediction of the input image $f_{w}(x)$ to $\beta_1$ and $\beta_2$, and thus not calibrated. Since $R$ is used to measure the robustness of NR-IQA models, different NR-IQA models for different input images should have the same original performance under no perceptual attack.
>
> **Regarding 1- SSIM as the distortion metric**: Thanks for the careful reading. In our implementation, we indeed use 1-SSIM as the $D(\cdot)$. We will clarify it in the text.

---

> > ### Author Response · Authors · 2022-07-28
> > **Responses to Reviewer Hmqo: (2)**
> >
> > **Regarding the basis for choosing these 6 images and their corresponding distortions**: We selected the 6 images from the test set of the LIVE dataset, the very first ``large-scale’’ human-rated IQA dataset. The four distortions (i.e., JPEG compression, JPEG2000 compression, Gaussian blur, and white noise) are the most basic distortions in the field of IQA, whose characterization and quantification have been widely regarded as already solved problems. According to our experimental results, this is in fact not the case, which suggests us to rethink the robust IQA problem from the most basic setting. In addition, we ensure that the selected images cover a broad quality range with discriminable perceptual quality.

---

### Official Review · Reviewer_qeax · 2022-07-10

**Rating:** 6
**Confidence:** 2
**Soundness:** 3 good
**Presentation:** 3 good
**Contribution:** 3 good

**Summary:**

This work presents a detailed analysis of the vulnerability of commonly used no-reference image quality assessment (NR-IQA) metrics to perceptual attacks. To this end, the authors include four NR-IQA metrics, BRISQUE, CORNIA, Ma19, and UNIQUE, into their analysis. The possibility of perceptual attacks is explored with four full reference IQA metrics, Chebyshev distance, SSIM, LPIPS, and DISTS. Based on the study, the authors have come up with several important observations that shed light on the design flows of the specific NR-IQA methods.

**Questions:**

1. Based on the experimental observations, can the authors provide some suggestions on some easy tricks that one could adopt to mitigate the vulnerability of perceptual quality metrics?

2. What about generating attacks that could simultaneously affect multiple NR-IQA methods? If that is not possible, can one rely on multiple metrics and hope for improved/complete resistance to perceptual attacks?


**Limitations:**

Has been addressed to a satisfactory level.

**Strengths And Weaknesses:**

Strengths

- First of its kind work to study the perceptual attack vulnerability of multiple NR-IQA methods
- The findings in the works point to the design flows of the existing NR-IQA methods while also giving insights into the need for developing NR-IQA methods that can well resist perceptual attacks.

Weaknesses

- The possibility of attacks that can simultaneously affect multiple NR-IQA metrics is not investigated

---

> ### Author Response · Authors · 2022-07-28
> **Responses to Reviewer qeax**
>
> Thanks for the valuable comments, pointing out current portions that need to be further polished and future directions that are worth exploring. Point-for-point comments are listed below.
>
> **Regarding suggestions to improve the perceptual robustness**: Thanks for the valuable comment, which is indeed what we are currently exploring. One of the key observations is that the counterexamples generated from one NR-IQA model are not transferrable to falsify other models. As suggested by the reviewer, we may directly leverage model ensemble tricks to mitigate the vulnerability of perceptual quality metrics, but it turns out not a viable solution (see the response to the next comment).
> We have also conducted some preliminary experiments by following the common practice of using the method of Madry et al. for adversarially training the UNIQUE model. Counterexamples during training are generated under an $\ell_{\infty}$-constraint with a radius of $\frac{4}{255}$, and the number of attacking iterations and the step size are set to $5$ and $\frac{2}{255}$, respectively. After training, we test the adversarially trained model using the proposed perceptual attack, and conduct an informal psychophysical experiment to verify the perceptual imperceptibility of the found counterexamples (by two of the authors). As shown in the table below, vanilla adversarial training significantly improves the robustness of the UNIQUE model. As suggested by the reviewer, in the future, we will delve deep into the design and training of perceptually robust NR-IQA models with improved effectiveness and efficiency.
>
> Table R4 SRCC and average ratio results of the UNIQUE model after the standard and adversarial training.
> | Standard Training |        | Adversarial Training |        |
> |:-----------------:|:------:|:--------------------:|:------:|
> |        SRCC       |    R   |         SRCC         |    R   |
> |      -0.0994      | 0.2059 |        0.6382        | 1.6395 |
>
> **Regarding one perturbation to fail multiple NR-IQA models**: Thanks for the excellent suggestion. Identification of one perceptually imperceptible perturbation to fail multiple NR-IQA models is indeed worth exploring, especially under the current circumstance that the counterexamples are not transferable (i.e., performing poorly in inter-model attacks). Conceptually, it is straightforward to extend the current computational method to look for such universal perturbation by replacing the second term in Eq. (2) with the average of discrepancy over $N$ NR-IQA models:
>
> $ {x}^{\star}=\underset{x}{\operatorname{argmax}} \- D(x, x_{0}) + \frac{\lambda}{N} \sum_{i=1}^{N} (f_{w}^{(i)}(x) - f(x_0))^2 $.
>
> Preliminary results indicate that it is possible to find one perturbation to fail multiple NR-IQA models in question by solving the above optimization problem with the proposed perceptual attack with human-in-the-loop. We kindly refer the reviewer to the appendix for the visual and numerical results.

---

### Official Review · Reviewer_1irv · 2022-07-12

**Rating:** 8
**Confidence:** 5
**Soundness:** 4 excellent
**Presentation:** 4 excellent
**Contribution:** 4 excellent

**Summary:**

This paper builds upon work from several fields (adversarial attacks, MAD competition, and Eigen-distortion analysis) which are analysis by synthesis methods used to generate small magnitude perturbations (by some pixel measure) that cause large change in model response (either in classification for adversarial attacks, or perceptual distance in MAD and Eigen-distortion).  The paper extends this to the field of no-reference image quality metrics, and compares their contribution with these other methods.  They successfully use this method to synthesize images that change the NR-IQA quality score significantly but are below human detection (with humans in the loop).  They use these images to quantify model to model performance, but also to elucidate how the construction of specific models leads to specific failure cases, which demonstrates one of the key values of analysis-by-synthesis.

**Questions:**

Table 1 is incomprehensible on first read and should be reorganized for more clarity.  It shouldn't require reading several pages past the table to fully digest what it is communicating.

Does the performance of UNIQUE actually improve in the face of the Brisque attacks over naive performance? (Same for Cornia under Unique attacks). What does that imply?

Why did the authors retrain the networks (at all), and why each of them on different datasets?

In addition to the four FR-IQA models, it would be interesting to compare to pixel MSE as an isotropic (in the pixel space) baseline. This will tell you the most about the NR-IQA model in the specific (as all pixel directs are equally penalized), whereas the other results show you a combination of the effects of the FR-IQA model used as well as the NR-IQA model, which is still useful but requires more effort to properly interpret.

More details about the exact lux of the illumination and the brightness of the displayed images on the displayed monitor would be useful.

For your derived value R, is there a theoretical upper bound that would provide satisfactory robustness?

Not a question, but this is a perfect encapsulation of the power of analysis by synthesis :
"The primary observation is that the difference in perturbations provides useful clues on how they extract quality-aware features."  The results presented in the final figure are insightful and very interesting to ponder.


**Limitations:**

Yes the author's carefully place their work in the context of many other works and carefully point on the limitations and advantages of their contribution with respect to this work.

**Strengths And Weaknesses:**

This is a very strong paper that builds nicely on several previous works, but meaningfully advances the work and extends it to an application that was not previously accessible. They also derive key insights from application of their method that could lead to followup work to improve NR-IQA models. This paper fits nicely within the NeuRIPS audience sweet spot, mixing knowledge and techniques from perceptual science, signal and image processing and machine learning.

There are a few details of the human experiments that should be elucidated further, these will be included in questions below.  I would encourage the authors to work in luminances when quantifying perceptual quality (in human experiments and in IQA models) and not fixed code values which are less generalizable to different displays and viewing environments.

---

> ### Author Response · Authors · 2022-07-28
> **Responses to Reviewer 1irv: (1)**
>
> Thanks for recognizing the merits of our work and for the confident positive recommendation. Point-for-point responses are listed below.
>
> **Regarding reorganizing Table 1 for more clarity**: Thanks for the comment. As suggested by the reviewer, we have divided the current Table 1 into two tables, one for intra-model attacks (i.e., counterexamples spotted to falsify the corresponding NR-IQA model by solving Problem (2)) and the other for inter-model attacks (i.e., counterexamples spotted to falsify other irrelevant NR-IQA models), as shown below. We have also rephrased the caption to make it much clearer.
>
> Table R1 SRCC and average ratio results of NR-IQA models under intra-model attacks.
> |  Metric |    SRCC   |   |   |   |     R     |       |       |       |
> |:-------:|:---------:|:-------:|:-------:|:-------:|:---------:|:------:|:------:|:------:|
> |  FR-IQA | Chebyshev |   SSIM  |  LPIPS  |  DISTS  | Chebyshev |  SSIM  |  LPIPS |  DISTS |
> | BRISQUE |   0.0959  |  0.2023 |  0.1221 |  0.1308 |   0.4116  | 0.3824 | 0.4412 | 0.7392 |
> |  CORNIA |   0.1998  |  0.0663 |  0.2092 |  0.0139 |   0.9141  | 0.4181 | 0.3682 | 0.2197 |
> |   Ma19  |   0.2441  |  0.1447 |  0.0959 |  0.2005 |   0.6188  | 0.3914 | 0.3943 | 0.3828 |
> |  UNIQUE |  -0.0994  | -0.1064 | -0.1813 | -0.1726 |   0.2059  | 0.1539 | 0.0927 | 0.1229 |
>
> Table R2 SRCC and average ratio results of NR-IQA models under inter-model attacks. The number in the parentheses shows the original performance of the NR-IQA model.
> |   Metric  |         SRCC        |        |        |        SRCC        |        |        |
> |:---------:|:-------------------:|:------:|:------:|:------------------:|:------:|:------:|
> |   Attack  |   BRISQUE (0.9231)  |        |        |   CORNIA(0.9650)   |        |        |
> |   Under Attack  |        CORNIA       |  Ma19  | UNIQUE |       BRISQUE      |  Ma19  | UNIQUE |
> | Chebyshev |        0.8230       | 0.6103 | 0.7777 |       0.8596       | 0.9730 | 0.9660 |
> |    SSIM   |        0.8684       | 0.9091 | 0.9276 |       0.9538       | 0.9790 | 0.9555 |
> |   LPIPS   |        0.7725       | 0.8806 | 0.8806 |       0.9154       | 0.9451 | 0.9451 |
> |   DISTS   |        0.7254       | 0.8806 | 0.8806 |       0.8997       | 0.9660 | 0.9625 |
> |   Metric  |         SRCC        |        |        |        SRCC        |        |        |
> |   Attack  |    Ma19 (0.9091)    |        |        |   UNIQUE(0.9301)   |        |        |
> |   Under Attack  |       BRISQUE       | CORNIA | UNIQUE |       BRISQUE      | CORNIA |  Ma19  |
> | Chebyshev |        0.7725       | 0.8387 | 0.8422 |       0.9381       | 0.9224 | 0.9242 |
> |    SSIM   |        0.8771       | 0.9102 | 0.9137 |       0.9346       | 0.9137 | 0.9276 |
> |   LPIPS   |        0.8718       | 0.8387 | 0.8893 |       0.9276       | 0.9346 | 0.9346 |
> |   DISTS   |        0.8387       | 0.8422 | 0.8858 |       0.9276       | 0.8945 | 0.9259 |
> |   Metric  |          R          |        |        |          R         |        |        |
> |   Attack  | BRISQUE ($+\infty$) |        |        | CORNIA ($+\infty$) |        |        |
> |   Under Attack  |        CORNIA       |  Ma19  | UNIQUE |       BRISQUE      |  Ma19  | UNIQUE |
> | Chebyshev |        2.1064       | 2.7651 | 2.4192 |       3.2428       | 4.3930 | 4.2005 |
> |    SSIM   |        2.4097       | 3.1472 | 3.2371 |       3.3378       | 4.1283 | 4.2050 |
> |   LPIPS   |        1.7400       | 3.1070 | 3.0490 |       3.5316       | 4.2963 | 4.6841 |
> |   DISTS   |        1.8017       | 3.1368 | 2.9193 |       3.8798       | 3.8258 | 4.0432 |
> |   Metric  |          R          |        |        |          R         |        |        |
> |   Attack  |   Ma19 ($+\infty$)  |        |        | UNIQUE ($+\infty$) |        |        |
> |   Under Attack |       BRISQUE       | CORNIA | UNIQUE |       BRISQUE      | CORNIA |  Ma19  |
> | Chebyshev |        2.4957       | 3.0544 | 3.1594 |       4.0762       | 3.6509 | 3.9139 |
> |    SSIM   |        3.6393       | 4.3902 | 4.4728 |       4.6850       | 2.9966 | 4.2564 |
> |   LPIPS   |        2.9033       | 3.0309 | 3.5103 |       4.4247       | 3.2835 | 4.9144 |
> |   DISTS   |        3.6089       | 3.4737 | 3.5533 |       4.8580       | 2.6109 | 4.1454 |
>
> **Regarding the performance improvement of UNIQUE in the face of BRISQUE attacks (or CORNIA under UNIQUE attacks)**: Thanks for the careful inspection of the current Table 1. We have taken a closer look at the slight performance improvement of UNIQUE under BRISQUE attacks. We find that this improvement arises because UNIQUE happens to correctly predict the relative orders of some perturbed images of very similar quality, in contrast to the incorrectly predicted orders of the corresponding initial images, all with very close quality prediction scores.

---

> > ### Author Response · Authors · 2022-07-28
> > **Responses to Reviewer 1irv: (2)**
> >
> > **Regarding retraining the networks on different datasets**: Thanks for the comment. The primary reason of re-training the four BIQA models is for the purpose of fair comparison. For BRISQUE and CORNIA, we re-trained them using the exact same training/validation/test splitting of the LIVE database. For Ma19 as an opinion-unaware BIQA model with its own training set specification, we retrained them using the same pristine-quality images in the original manuscript, but considered only four types of distortions overlapping with LIVE, i.e., white Gaussian noise, Gaussian blur, JPEG compression, and JPEG2000 compression. As for UNIQUE that enables training on multiple IQA datasets without perceptual scale realignment, we followed the training protocols in the original manuscript, and combined six IQA datasets as the training set. This also allows us to investigate whether incorporating more human-rated quality data improves the perceptual robustness.
> >
> > **Regarding adding pixel MSE as an isotropic baseline**: Thanks for the suggestion. We have actually adopted the MSE in the pixel domain as the perceptual constraint. In the below table, we compare DISTS against MSE, from which we find consistently that a more perceptual FR-IQA model generally delivers more effective attacks. Due to the space limit and the widely adoption of the Chebyshev distance in the adversarial machine learning community, we choose only to show the Chebyshev results. Nevertheless, we are comfortable to instead present the MSE results.
> >
> > Table R3 SRCC and average ratio results using MSE and DISTS as the perceptual constraints.
> > | FR-IQA | Metric | BRISQUE | CORNIA |  Ma19  |  UNIQUE |
> > |:------:|:------:|:-------:|:------:|:------:|:-------:|
> > |   MSE  |  SRCC  |  0.2128 | 0.2934 | 0.2568 |  0.1779 |
> > |        |    R   |  0.4616 | 1.7028 | 0.7273 |  0.3687 |
> > |  DISTS |  SRCC  |  0.1308 | 0.0139 | 0.2005 | -0.1726 |
> > |        |    R   |  0.7392 | 0.2197 | 0.3828 |  0.1229 |
> >
> > **Regarding the illumination and the brightness of the displayed images**: Thanks for the comment. The subjective experiment was carried out in an indoor office environment with a normal lighting condition (with approximately 200 lux) and without reflecting ceiling walls and floors. The peak luminance of the displayed images is mapped to 200 cd/m^2. These are suggested in the BT. 500 recommendations.
> >
> > **Regarding the average ratio $R$**: The definition of $R$ is mainly inspired by the Peak Signal-to-Noise Ratio (PSNR), which is widely used in signal and image processing. By definition, an NR-IQA model with perfect perceptual robustness will make exactly the same prediction for an initial image $x$ and the corresponding counterexample $x^{\star}$, correct or not. In that case, the denominator of Eq. (6) $\| f_{w}(x) – f_{w}(x^{\star}) \|$ is zero, and the value of $R$ would be positive infinity. And we admit that it is difficult to obtain a theoretical (and universal) lower bound, above which the perceptual robustness is guaranteed. We conjecture that such a bound may be dependent on image content as well as distortion type and level.
> >
> > **Regarding the final image**: Thanks for appreciating the drawing of the Fig. 3.

---

### Meta-Review · Area_Chair_r5ka · 2022-08-24

**Recommendation:** Accept
**Confidence:** Certain

**Metareview:**

All the reviewers are in agreement in their recommendations to accept the paper. The topic of the paper brings along many different sub-fields, including adversarial attacks and image quality assessment, and should be interesting to several folks in the community. There are several constructive comments by the reviewers that I’d encourage the authors to address, especially the ones asking for clearly state the motivation of this line of research, as well synthesis of ideas from the study that would enable ideas for better defense against such attacks.

**Award:**

No

---

### Decision · Program_Chairs · 2022-09-14

Accept